# MOLECULAR GRAPH ENHANCED TRANSFORMER FOR RETROSYNTHESIS PREDICTION

## ABSTRACT

With massive possible synthetic routes in chemistry, retrosynthesis prediction is still a challenge for researchers. Recently, retrosynthesis prediction is formulated as a Machine Translation (MT) task. Namely, since each molecule can be represented as a Simplified Molecular-Input Line-Entry System (SMILES) string, the process of synthesis is analogized to a process of language translation from reactants to products. However, the MT models that applied on SMILES data usually ignore the information of natural atomic connections and the topology of molecules. In this paper, we propose a Graph Enhanced Transformer (GET) framework, which adopts both the sequential and graphical information of molecules. Four different GET designs are proposed, which fuse the SMILES representations with atom embedding learned from our improved Graph Neural Network (GNN). Empirical results show that our model significantly outperforms the Transformer model in test accuracy.

## 1 INTRODUCTION

Retrosynthesis prediction aims to predict a set of suitable reactants that can synthesize the desired molecule via a series of reactions. It pushes forward an immense influence in agriculture, medical treatment, drug discovery and so on. However, the retrosynthesis prediction is challenging since there are massive possible synthetic routes available and it is often difficult to navigate the direction of retrosynthesis process. Indeed, each bond in the target molecule may represent a possible retrosynthetic disconnection, leading to a vast space of possible starting materials. Besides, the difference between two synthetic routes may be subtle, which usually depends on the global structures. Actually, planning a proper retrosynthetic route for a complex molecule is also a tough work even for the professional chemists.

One of the prevailing methods is to deem the retrosynthesis prediction as a machine translation task. This analogy is comprehensible since every molecule has a unique text representation named SMILES (Weininger, 1988). In this case, given a target molecule written in SMILES notation, the retrosynthesis prediction is just to predict a string of SMILES which represents the reactants. Based on this idea, Liu et al. (2017) first applied the LSTM with attention mechanism in retrosynthesis prediction and achieved comparable performance compared with previous traditional methods. Whereafter, many works (Karpov et al., 2019; Zheng et al., 2019; Lin et al., 2019; Lee et al., 2019) tried to employ Transformer (Vaswani et al., 2017), which is a more powerful Sequence-to-Sequence(Seq2Seq) model, to improve prediction accuracy in retrosynthesis. However, these methods just utilize the sequential representations of the molecule, while ignoring the natural topological connections between atoms within the molecule. These atomic connections can provide more flexible and accurate chemical information, which is critical in many related chemical tasks like molecular representation (Duvenaud et al., 2015; Gilmer et al., 2017) and chemical reaction prediction (Jin et al., 2017; Do et al., 2019). We believe that the absence of this molecular graph information hinders the further improvement of the present methods for retrosynthesis. How to effectively make use of this natural graphical information of the molecular structure, therefore, becomes a vital problem.

To tackle this problem, we propose Graph Enhanced Transformer(GET) framework that can enjoy the advantage of both graph-level representations and sequence-level representations. Specifically, to solve the retrosynthesis problem, we design an improved Graph Neural Network(GNN) called Graph Attention with Edge and Skip-connection (GAES) to learn each atom's representation, and

try four strategies to incorporate it with the original SMILES representation in the encoder. The main contributions of this paper are as follows:

- We propose a new framework called GET that fuses graphical representations with sequential representations of the target molecule to solve retrosynthesis prediction task.
- We design a powerful GNN called GAES that learns high-quality representations of atom nodes in a self-attention manner with bond features, and it is less affected by the side-effect of stacking more layers.

GET is evaluated on USPTO-50K, a common benchmark dataset for retrosynthesis. Experimental results show that our model achieves new records for top-1 prediction accuracy in the state-of-the-art methods and outperforms Seq2Seq-based methods in all tested top-$n$ accuracy, demonstrating the effectiveness of fusing the molecular graph information with the SMILES sequence information.

## 2 RELATED WORK

Prior work on retrosynthesis can be mainly summarized into two categories: template-based methods and template-free methods.

**Template-based Methods** The majority of computer-aided retrosynthetic methods in the early period were relied on encoding reaction templates or generalized subgraph matching rules. LHASA (Corey & Wipke, 1969) was the first software for retrosynthetic analysis. Recently, one of the most well-known retrosynthesis analysis tool is Synthia (Szymkuć et al., 2016) that integrated about 70,000 hand-encoded reaction rules collected by manual. Based on the 60,000 reaction templates derived from 12 million single-step reaction examples, Schreck et al. (2019) introduced Reinforcement Learning (RL) into this area by treating retrosynthesis as a game whose goal is to identify policies that make (near) optimal reaction choices during each step of retrosynthetic planning. Segler et al. (2018) extracted two sets of transformation rules and combined Monte Carlo tree search with symbolic AI to discover possible retrosynthetic routes. Besides manual extracted rules, some works (Law et al., 2009; Segler & Waller, 2017; Coley et al., 2017) tried to collect reaction templates automatically and perform retrosynthesis based on these automated templates. Although template-based methods work well in many cases, they still face a serious drawback that they generally cannot achieve accurate prediction accuracy outside of their known templates.

**Template-free Methods** Our model just belongs to this category. Emerging template-free methods are to treat retrosynthesis as a machine translation task as introduced in section 1. Since these methods do not need any reaction templates and prior chemistry knowledge, they are attracting more and more attention from academia. Moreover, without the constraint of fixed templates, they have the potential of discovering novel synthetic routes. The most related work to ours are Zheng et al. (2019) Lin et al. (2019); Karpov et al. (2019); Lee et al. (2019) that apply Transformer in retrosynthesis prediction.

## 3 BACKGROUND

In this section, we first introduce how GNN is used in learning the molecule (or atoms) representation, and then describe how the Transformer model is previously applied in retrosynthesis prediction.

### 3.1 GNN FOR MOLECULE REPRESENTATION LEARNING

GNN has been widely used in learning the representation of the molecule and its atoms. Naturally, molecules can be represented as graph structure with atoms as nodes and bonds as edges. Suppose that a molecular graph $G$ has initial node representations $\boldsymbol{h}_v$ and edge representations $\boldsymbol{e}_{vw}$, a typical one-layer GNN can learn new and more powerful node representations from $G$ by the following message passing process described in Gilmer et al. (2017):

$$\boldsymbol{m}_v = \sum_{w \in N(v)} M\left(\boldsymbol{h}_v, \boldsymbol{h}_w, \boldsymbol{e}_{vw}\right), \tag{1}$$

$$\boldsymbol{h}_v^{new} = U\left(\boldsymbol{h}_v, \boldsymbol{m}_v\right), \tag{2}$$

where $N(v)$ denotes the neighbors of node $v$ in graph $G$, $M$ is the message function that is responsible for collecting information from neighbors, and $U$ is the update function for fusing collected

information $\boldsymbol{m}_v$ with old node representation $\boldsymbol{h}_v$ to obtain the new node representation $\boldsymbol{h}_v^{new}$. Further, we can stack several these GNN layers to capture higher-order neighbors' information.

Then a readout function $R$ can be used to integrate all node representations into a whole graph representation $\boldsymbol{g}$:

$$\boldsymbol{g} = R(\{\boldsymbol{h}_v | v \in G\}). \tag{3}$$

The $\boldsymbol{h}_v$ and $\boldsymbol{g}$, which represent the atoms and the whole molecule, are often trained in an end-to-end way for a specific chemical task, such as chemical properties prediction, reaction prediction and molecule optimization.

## 3.2 TRANSFORMER FOR RETROSYNTHESIS PREDICTION

Transformer (Vaswani et al., 2017) is a Seq2Seq model that has shown excellent performance in machine translation task. Also, it has been applied in chemical reaction prediction and retrosynthesis prediction before. Given an input SMILES that represents the target molecule and a specified reaction type (optional), retrosynthesis prediction is to predict the output SMILES which represents the possible reactants that can synthesis the target molecule in the specified reaction type. Thus, retrosynthesis prediction can be deemed as a machine translation task whose source language is target molecule SMILES and the target language is reactants SMILES.

In this view, Transformer can be applied to retrosynthesis prediction as the same as to machine translation. Since Transformer is a mature model that has been widely used in natural language processing (NLP), we just give a simple introduction here. Specifically, Transformer follows an encoder-decoder structure and is composed of several combinations of multi-head attention layers and position-wise feed forward layers. The encoder consists of a stack of $N = 6$ identical layers. Each layer includes two main components: (multi-head) self-attention layer and feed-forward network. Given an input vectors $(\boldsymbol{p}_1, ..., \boldsymbol{p}_n)$, $\boldsymbol{p} \in \mathbb{R}^d$, the $t$-th output $\boldsymbol{s}_t$ of the self-attention layer is calculated by:

$$\boldsymbol{q}_t = \boldsymbol{W}_q \boldsymbol{p}_t, \quad \boldsymbol{k}_m = \boldsymbol{W}_m \boldsymbol{p}_m, \quad \boldsymbol{v}_t = \boldsymbol{W}_v \boldsymbol{p}_t, \quad \boldsymbol{s}_t = \sum_{m=1}^{n} \text{softmax}(\frac{<\boldsymbol{q}_t, \boldsymbol{k}_m>}{\sqrt{d_k}})\boldsymbol{v}_t, \tag{4}$$

where $d_k$ is the dimension of $\boldsymbol{q}$ and $\boldsymbol{k}$, $\boldsymbol{W}_q, \boldsymbol{W}_m, \boldsymbol{W}_v$ are weight matrices. One such operation is called one head, and we can concatenate several heads to change to multi-head self attention.

The feed-forward network is composed of two linear transformations with a ReLU activation:

$$\text{FFN}(\boldsymbol{x}) = \max(0, \boldsymbol{x}\boldsymbol{W}_1 + \boldsymbol{b}_1)\boldsymbol{W}_2 + \boldsymbol{b}_2, \tag{5}$$

where $\boldsymbol{W}_1, \boldsymbol{W}_2$ are weight matrices and $\boldsymbol{b}_1, \boldsymbol{b}_2$ are biases.

Similarly, the decoder is also mainly composed of multi-attention layers and position-wise feed forward layers. It will generate the output SMILES step by step. At step $t$, it utilizes the encoder's output $(\boldsymbol{p}_1', ..., \boldsymbol{p}_n')$ and all previous steps' output $(x_1', ..., x_{t-1}')$ to generate the next SMILES character $x_t'$. This process repeated until generating a specific termination character, i.e., $x_t' = <EOS>$.

## 4 GRAPH ENHANCED TRANSFORMER

In this section, we provide the details about our Graph Enhanced Transformer (GET) framework for retrosynthesis prediction. Figure 1 shows an overview of GET. On the whole, GET is accord with typical encoder-decoder structure, of which the integral encoder is composed of the graph encoder and transformer encoder for learning the representation in graph-level and sequence-level respectively. Given a target molecule's SMILES, it will first pass through the two encoders somehow to get the hidden representation of each character, and then the decoder will utilize these hidden representations to generate the output SMILES.

## 4.1 GRAPH ENCODER

We design a new powerful GNN called Graph Attention with Edge and Skip-connection (GAES) as the graph encoder, which can learn the representation of each atom in a molecule. This graphical

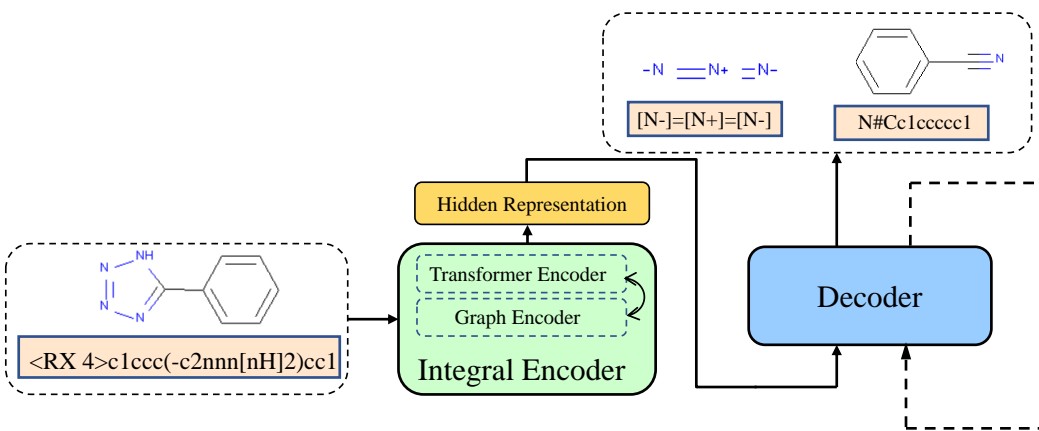

Figure 1: Overview of GET. The input SMILES will be processed by the two-sub encoders (graph encoder and transformer encoder) somehow to be transformed to its hidden representation. Then, at each step, the decoder will utilize the hidden representation and all outputs of the previous steps to generate the present step's SMILES character.

representation reflects the connection of atoms within a molecule and may play a significant role in further alleviating the long-term dependency problem to avoid generating chemically invalid output. We use RDKit(Landrum, 2016) to transform a SMILES into the molecular graph, whose nodes are atoms and edges are chemical bonds. The input representation of the atom node is a 21-dimensional vector that contains some chemical information about the atom, and the detail can be found in Table 1 (nearly consistent with Gilmer et al. (2017)). The input representation of the edge is a 4-dimensional one-hot vector that encodes the bond types including single, double, triple and aromatic.

Since the SMILES sequence $(x_1, ..., x_n)$ has been transformed into a graph $G$ with the input representations $(\boldsymbol{h}_1, ..., \boldsymbol{h}_N)$ for nodes and $\{\boldsymbol{e}_{ij}\}$ for edges that exist between node $i$ and node $j$, our GNN will produce new representation $\boldsymbol{h}'_i$ for each node $i$ by the following message passing operations:

$$\alpha_{ij} = \frac{\exp\left(\text{LeakyReLU}\left(\mathbf{a}^T\left[\boldsymbol{W}\boldsymbol{h}_i\|\boldsymbol{W}\boldsymbol{h}_j\|\boldsymbol{e}_{ij}\right]\right)\right)}{\sum_{k\in\mathcal{N}_i}\exp\left(\text{LeakyReLU}\left(\mathbf{a}^T\left[\boldsymbol{W}\boldsymbol{h}_i\|\boldsymbol{W}\boldsymbol{h}_k\|\boldsymbol{e}_{ik}\right]\right)\right)}, \tag{6}$$

$$\hat{\boldsymbol{h}}_i = \sigma\left(\sum_{j\in\mathcal{N}_i}\alpha_{ij}\boldsymbol{W}\boldsymbol{h}_j\right), \tag{7}$$

where $\mathbf{a} \in \mathbb{R}^{(2F'+E)}$ is a weight vector for attention mechanism and $\boldsymbol{W} \in \mathbb{R}^{F'\times F}$ is a weight matrix for transforming the node features, so $F$ is the input dimension of nodes, $F'$ is the output dimension of nodes and $E$ is the dimension of edges. $\mathcal{N}_i$ is the set of first-order neighbors of node $i$ (including itself). $\sigma$ is an activation function, e.g., ReLU.

In practice, we perform $K$ multi-head attention (Veličković et al., 2017) to enrich the model capacity and to stabilize the learning process. Each attention head has its own parameters and we average their outputs to get better representation:

$$\hat{\boldsymbol{h}}_i = \sigma\left(\frac{1}{K}\sum_{k=1}^{K}\sum_{j\in\mathcal{N}_i}\alpha_{ij}^k\boldsymbol{W}^k\boldsymbol{h}_j\right). \tag{8}$$

The above operations can be seen as GAT (Veličković et al., 2017) extended to include edge features. Then, to mitigate the accuracy reduction issue (Kipf & Welling, 2016) caused by stacked graph convolution layers, we adopt the gated skip-connection mechanism (Ryu et al., 2018) to get the final representation:

$$\boldsymbol{z}_i = \text{sigmod}(\boldsymbol{U}_1\hat{\boldsymbol{h}}_i + \boldsymbol{U}_2\boldsymbol{h}_i + \boldsymbol{b}), \tag{9}$$

$$\boldsymbol{h}'_i = \boldsymbol{z}_i \odot \hat{\boldsymbol{h}}_i + (1 - \boldsymbol{z}_i) \odot \boldsymbol{h}_i, \tag{10}$$

Table 1: Input representation of atom nodes

| Atom Feature | Description |
| --- | --- |
| Atom type | C, N, O, S, P, B, F, I, Sn, Cl, Br, Se, Si (one-hot) |
| Atom number | Numbers of protons (integer) |
| Acceptor | Accepts electrons (binary) |
| Donor | Donates electrons (binary) |
| Aromatic | In an aromatic system (binary) |
| Hybridization | sp, sp2, sp3 (one-hot or null) |
| Number of Hydrogens | (integer) |

where $\boldsymbol{U}_1$, $\boldsymbol{U}_2$ and $\boldsymbol{b}$ are trainable parameters.

Note that the above operations are just in one layer of our GAES, and we can stack several layers to capture the information about higher-order atom neighbors so that to obtain more comprehensive representations.

## 4.2 TRANSFORMER ENCODER

The transformer encoder is the same as described in section 3, which can capture the sequential representations of molecules (or atoms) represented by SMILES. The original SMILES $(x_1, ..., x_n)$ is changed to a sequence of vectors $(\boldsymbol{p}_1, ..., \boldsymbol{p}_n), \boldsymbol{p} \in \mathbb{R}^d$ after passing the embedding layer. And it will be further updated to vectors $(\boldsymbol{p}'_1, ..., \boldsymbol{p}'_n)$ by the transformer encoder.

## 4.3 REPRESENTATION FUSION

Intuitively, graphical representations reflect the intrinsic structural features of molecules and should be beneficial to generate chemical-valid and more accurate SMILES output. To this end, we propose four fusion strategies to fuse these graphical and sequential embeddings.

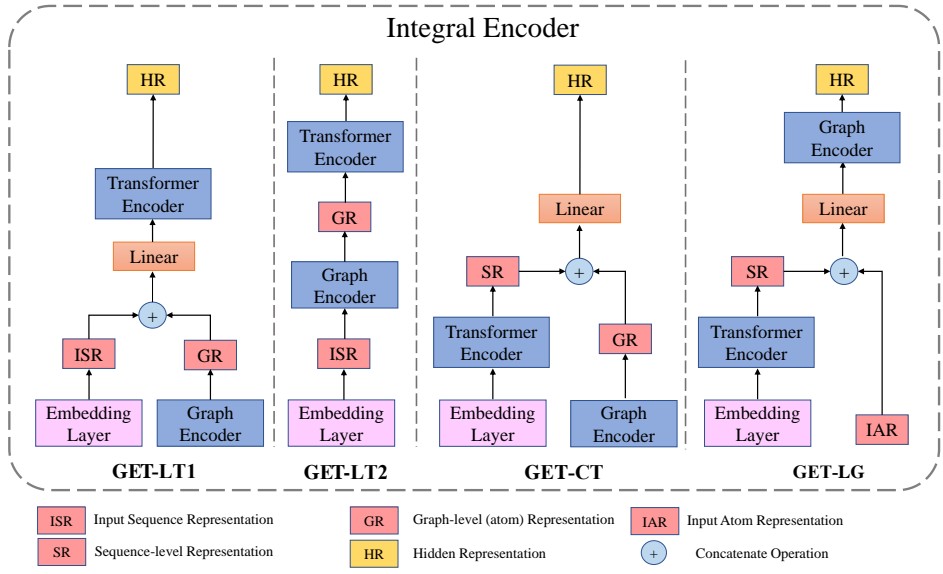

Figure 2: Illustration of four fusion strategies in the encoder of GET. The integral encoder is composed of two sub-encoders: graph encoder and transformer encoder. The embedding layer is as described in 4.2 . The hidden representation of the target molecule can be obtained in four ways (GET-LT1, GET-LT2, GET-CT and GET-LG).

### 4.3.1 GRAPH LINK TRANSFORMER

As shown in Figure 2 (GET-LT1), we concatenate the atom representations with embeddings of SMILES and perform a linear transformation by weight matrix $M$. Then the new representations are sent to the transformer encoder to produce the output of the integral encoder. For non-atomic characters in SMILES, the corresponding atom representations are set to zero vector. Formally,

$$\hat{\boldsymbol{p}}_i = [\boldsymbol{p}_i \| \boldsymbol{h}_i'], \tag{11}$$
$$(\boldsymbol{v}_1, ..., \boldsymbol{v}_n) = (\boldsymbol{p}_1', ..., \boldsymbol{p}_n') = \text{TransformerEncoder}(M\hat{\boldsymbol{p}}_1, ..., M\hat{\boldsymbol{p}}_n), \tag{12}$$

where $\boldsymbol{h}_i' = \boldsymbol{0}$ if $x_i$ is non-atomic character, $\|$ is the concatenation operation. $(\boldsymbol{v}_1, ..., \boldsymbol{v}_n)$ is integral encoder's output.

Considering that there may exist inconsistency between encoder and decoder since the initial atom features inputted to the graph encoder cannot be utilized by the decoder when making inference, we try another way by replacing the natural atom features $(\boldsymbol{h}_1, ..., \boldsymbol{h}_n)$ with $(\boldsymbol{p}_1, ..., \boldsymbol{p}_n)$ as the input representations of the graph encoder:

$$(\boldsymbol{h}_1', ..., \boldsymbol{h}_n') = \text{GraphEncoder}(\boldsymbol{p}_1, ..., \boldsymbol{p}_n). \tag{13}$$

In this way, the graph encoder can enhance the original sequential representations $(\boldsymbol{p}_1, ..., \boldsymbol{p}_n)$ with molecule structure information directly, but the natural atom features have to be "sacrificed'. Then we send the output of the graph encoder to the transformer encoder to get the final output:

$$(\boldsymbol{v}_1, ..., \boldsymbol{v}_n) = (\boldsymbol{p}_1', ..., \boldsymbol{p}_n') = \text{TransformerEncoder}(\boldsymbol{h}_1', ..., \boldsymbol{h}_n'). \tag{14}$$

We name the first scheme GET-LT1 and the second scheme GET-LT2.

### 4.3.2 GRAPH CONCATENATE WITH TRANSFORMER

As shown in Figure 2 (GET-CT), we concatenate the outputs of the graph encoder and transform encoder, and also perform linear transformation to get the output of the integral encoder:

$$\hat{\boldsymbol{p}}_i = [\boldsymbol{p}_i' \| \boldsymbol{h}_i'], \tag{15}$$
$$(\boldsymbol{v}_1, ..., \boldsymbol{v}_n) = (M\hat{\boldsymbol{p}}_1', ..., M\hat{\boldsymbol{p}}_n'). \tag{16}$$

The notations are consistent with 4.3.1.

### 4.3.3 TRANSFORMER LINK GRAPH

As shown in Figure 2 (GET-LG), the SMILES sequence first pass through the transformer encoder, then it is concatenated with natural atom features $(\boldsymbol{h}_1, ..., \boldsymbol{h}_n)$ to be the input representation of the graph encoder. Those non-atomic characters are added into the molecular graph as standalone nodes which do not connect with any other node, and their "atom feature vectors" are just zero vectors. Finally, the graph encoder's output will be the integral encoder's output:

$$(\boldsymbol{v}_1, ..., \boldsymbol{v}_n) = (\boldsymbol{h}_1', ..., \boldsymbol{h}_n') = \text{GraphEncoder}([\boldsymbol{p}_1' \| \boldsymbol{h}_1], ..., [\boldsymbol{p}_n' \| \boldsymbol{h}_n]). \tag{17}$$

### 4.4 DECODER

The decoder is the same as vanilla Transformer's (Vaswani et al., 2017) decoder which has been introduced in section 3. At step $t$, the encoder's output $(\boldsymbol{v}_1, ..., \boldsymbol{v}_n)$ and all previous steps' output $(x_1', ..., x_{t-1}')$ are used by the decoder to generate the next SMILES character $x_t'$ until $x_t' = <EOS>$.

## 5 EXPERIMENTS

In this section, we evaluate our model for retrosynthesis prediction on a common benchmark dataset USPTO-50K which is derived from USPTO granted patents that includes 50,033 reactions classified into 10 reaction types. A reaction is described as a pair of sequences which consist of SMILES notations for target molecule (with reaction type) and reactants. For example, an heterocycle formation reaction is described as: ("<RX_4>c1ccc(-c2nnn[nH]2)cc1", "N#Cc1ccccc1.[N-]=[N+]=[N-]"), where "<RX_4>" represents heterocycle formation reaction, "c1ccc(-c2nnn[nH]2)cc1" is SMILES of the target molecule, "N#Cc1ccccc1" and "[N-]=[N+]=[N-]" are SMILES of two reactants separated by ".".

### 5.1 SETTINGS

- **Dataset Split** Many previous works (Liu et al., 2017; Coley et al., 2017; Zheng et al., 2019; Karpov et al., 2019) follow a specific split strategy with 40,029, 5,004 and 5,004 reactions for training, validation and testing, and we keep the same.

- **Implementation** For the graph encoder, it is implemented based on DGL (Wang et al., 2018). We stack 3 identical layers in our GNN. The input and output dimension of nodes are set to 21 and 256 respectively. The number of multi-head is set to 2; For the transformer encoder and the decoder, we implement them using OpenNMT (Klein et al., 2017), and the parameter settings are presented in the code; Besides, the final dimension of the integral encoder's output $v$ is set to 256.

### 5.2 RESULT

We compare our model with the vanilla Transformer (Vaswani et al., 2017), Rule-based Expert System mentioned in Liu et al. (2017) , Similarity (Coley et al., 2017) and LSTM+Attention (Liu et al., 2017). Note that the results of vanilla Transformer are based on our own experiments since the results reported by previous works (Zheng et al., 2019; Lin et al., 2019; Karpov et al., 2019; Lee et al., 2019) are different from each other. The retrosynthesis prediction accuracy across all classes is provided in Table 2. Moreover, we also test the performance of GET-LT1 when removing the reaction type from the original dataset, and the result is shown in Table 3.

Table 2: Comparison of top-$n$ accuracies across all classes

| Model | top-$n$ accuracy (%) | | | |
|---|---|---|---|---|
| | 1 | 3 | 5 | 10 |
| Rule-based Expert System | 35.4 | 52.3 | 59.1 | 65.1 |
| LSTM+Attention | 37.4 | 52.4 | 57.0 | 61.7 |
| Similarity | 52.9 | **73.8** | **81.2** | **88.1** |
| Transformer (baseline) | 54.3 | 68.4 | 72.0 | 74.4 |
| GET-CT (our) | 55.9 | 70.1 | 73.2 | 76.3 |
| GET-LG (our) | 54.9 | 69.7 | 72.2 | 74.6 |
| GET-LT2 (our) | 56.2 | 69.4 | 72.5 | 74.7 |
| GET-LT1 (our) | **57.4** | 71.3* | 74.8* | 77.4* |

Table 3: Comparison of top-$n$ accuracies across all classes without reaction type

| Model | top-$n$ accuracy (%) | | | |
|---|---|---|---|---|
| | 1 | 3 | 5 | 10 |
| Similarity | 37.3 | 54.7 | 63.3 | 74.1 |
| Transformer (baseline) | 42.3 | 57.5 | 61.0 | 65.7 |
| GET-LT1 (our) | **44.9** | **58.8** | **62.4** | **65.9** |

Results show that our models outperform all of previous methods in top-1 accuracy, and our best model GET-LT1 achieves the new state-of-the-art among all Seq2Seq-based methods, i.e,

LSTM+Attention, vanilla Transformer and models of GET. Compared with vanilla Transformer, GET-LT1 can improve the prediction accuracy by 3.1%, 2.9%, 2.8% and 3.0% in top-1, top-3, top-5 and top-10 accuracy. Other variants also have varying degrees of performance improvement over vanilla Transformer. And our model can retain this comprehensive superiority after removing the reaction type, demonstrating that molecule structure information can help Transformer to predict more accurate reactants. Besides, note that Similarity (Coley et al., 2017) is a template-based model which predicts 100 candidates and just chooses the top-10 as the final result, while other template-free models, i.e., LSTM+Attention, Transformer and GET, are trained to accurately predict the top-1 output and only generate 10 candidates using beam search. Therefore, it is not surprising that Similarity achieves very high accuracy. Nevertheless, even in this unfair situation, our models can surpass Similarity up to 4.5% in top-1 accuracy.

In addition, we present the detailed top-10 accuracy of three Seq2Seq-based models (LSTM+Attention, vanilla Transformer and GET-LT1) for each reaction class in Table 4. Results show that our approach can improve vanilla Transformer on 9 of 10 reaction classes by a margin of 1.3% to 17.4%, indicating the better generalization ability and comprehensiveness of our model.

Table 4: Comparison of the top-10 accuracy for each reaction class

| Model | top-10 accuracy (%) | | | | | | | | | |
|---|---|---|---|---|---|---|---|---|---|---|
| | 1 | 2 | 3 | 4 | 5 | 6 | 7 | 8 | 9 | 10 |
| LSTM+Attention | 57.5 | 74.6 | 46.1 | 27.8 | 80.0 | 62.8 | 67.8 | 69.1 | 47.3 | 56.5 |
| Transformer (baseline) | 73.5 | 81.9 | 62.7 | 52.2 | 86.1 | 71.5 | 80.0 | **83.9** | 65.2 | 73.9 |
| GET-LT1 (our) | **76.6** | **84.2** | **66.1** | **65.6** | **89.2** | **75.7** | **81.3** | 81.5 | **71.7** | **91.3** |

Furthermore, the rate of producing grammatically invalid SMILES for different beam sizes are shown in Table 5 (with reaction type). As can be seen, after fusing molecule structure information, the model is more inclined to generate chemical-valid SMILES compared with vanilla Transformer, since the graphical representations, which directly capture the topological connection of atoms, are able to break the limitation of the SMILES sequence and can give the model additional guidance to produce chemical-valid compound.

Table 5: The rate of producing grammatically invalid SMILES for different beam sizes

| Model | invalid SMILES' rate (%) when beam size $k =$ | | | |
|---|---|---|---|---|
| | 1 | 3 | 5 | 10 |
| LSTM+Attention | 12.2 | 15.3 | **18.4** | **22.0** |
| Transformer (baseline) | 3.5 | 14.3 | 20.3 | 30.2 |
| GET-LT1 (our) | **2.2** | **13.4** | 19.5 | 29.3 |

## 6 CONCLUSION AND FUTURE WORK

We propose Graph Enhanced Transformer(GET), an effective framework that successfully combines the graphical and sequential representations of the molecule to improve the retrosynthesis prediction performance. Experiments indicate that our model outperforms state-of-the-art Seq2Seq-based methods on USPTO-50K dataset, and shows promising ability in reducing invalid SMILES rate.

In the future, we plan to 1) explore how to utilize the molecular graph information in the decoder. 2) research how to let the decoder generate reactants in the form of graph directly.

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
