# OpenReview forum: "Molecular Graph Enhanced Transformer for Retrosynthesis Prediction"
_ICLR.cc/2020/Conference — Reject_

### Official Review · AnonReviewer4 · 2019-10-19
**Official Blind Review #4**

**Rating:** 3

**Review:**


This paper focuses on the retrosynthesis prediction problem which to my understanding is the factorization of a target molecule into simpler structures. Previously, retrosynthesis prediction has been tackled as a language translation problem by using a Seq2Seq algorithm called "Transformer". This sequential approach was possible because molecules can be expressed as strings using the following format: Simplified Molecular-Input Line-Entry System (SMILES).

Despite its good performance, language translation methods ignore the graphical structure of molecules. This paper proposes to add a graph neural network in front of the Seq2Seq/Transformer network to exploit the graphical structure, hence the name “Graph Enhanced Transformer (GET)”. The Graph Neural Network considered in this work also uses an attention mechanism similarly to Graph Attention Networks.

The main contribution is the addition of a Graph Neural Network to a Seq2Seq model for retrosynthesis prediction which provides state of the art results.

From a machine learning / representation learning perspective, I do not consider the paper is innovative enough. Even so, the paper shows strong results for retrosynthesis prediction, I guess it would be a good fit in a chemistry conference.


Things to improve:

There are plenty of works related to graph Autoencoders and Graph generative models that are not mentioned in the related work. It would be great to dedicate a section for them. Here some examples:

- Li, Y., Vinyals, O., Dyer, C., Pascanu, R., & Battaglia, P. (2018). Learning deep generative models of graphs.

-Simonovsky, M., & Komodakis, N. (2018, October). Graphvae: Towards generation of small graphs using variational autoencoders.

-De Cao, N., & Kipf, T. (2018). MolGAN: An implicit generative model for small molecular graphs.

-You, J., Ying, R., Ren, X., Hamilton, W. L., & Leskovec, J. (2018). Graphrnn: Generating realistic graphs with deep auto-regressive models.

All experiments are compared to author’s re-implementations of other works. It would be interesting to directly report other work accuracies.


**Experience Assessment:**

I have published one or two papers in this area.

**Review Assessment: Checking Correctness Of Derivations And Theory:**

I assessed the sensibility of the derivations and theory.

**Review Assessment: Checking Correctness Of Experiments:**

I assessed the sensibility of the experiments.

**Review Assessment: Thoroughness In Paper Reading:**

I read the paper at least twice and used my best judgement in assessing the paper.

---

> ### Author Response · Authors · 2019-11-12
> **Author Response #4**
>
> Thank you for your review.
> (1) For retrosynthesis prediction,  our work is the first attempt to utilize both the graphical and sequential information of molecules.  We fully investigate four organization forms of graph neural network (GNN) with Transformer and significantly improve the prediction performance in many metrics, i.e.,  accuracy and chemical invalid rate. Besides, the GNN we used is also well-designed to further improve the performance, which is more powerful to learn the representation of atoms.
>
> (2) We also thank for your directional suggestion  (we will consider the chemistry conference).
>
> (3) The decoder of our model is vanilla Transformer's decoder, thus it's a sequence generative model. The molecular graph information is only used in the encoder phase, since we cannot get the whole molecular graphs of the reactants when decoding. However, how to directly generate a molecular graph not a SMILES sequence is just what we are working on, and we will add the related work about graph Autoencoders and Graph generative models later.  Thank you very much.
>
> (4) We report the result of Rule-based Expert System, LSTM+Attention and Similarity by their original paper, not our re-implementation. We just re-implement the Transformer model's result since there have been 4 papers using Transformer to solve retrosynthesis, while they reported different results. So we had to re-implement it by ourselves. Actually, their results are closed to ours, and our GET is also obviously better than them.

---

### Official Review · AnonReviewer3 · 2019-10-23
**Official Blind Review #3**

**Rating:** 3

**Review:**

~The authors propose an enhancement to the transformer architecture that takes molecule graph structure into account.~

I applaud the authors work on making more physically plausible machine learning constraints. However, I feel like this work is incremental and does not vastly improve SoA.

For all tables, what are the error bars on the accuracy?

There are multiple possible SMILES strings for any unique molecule (which includes the canonical SMILES string.) Does bootstrapping your Transformer model with multiple non-canonical SMILES for the same input molecules improve performance?

Many seq2seq molecules allow an attention mechanism on the input sequence while decoding, and that seems like this would be useful for this data. What would the impact of this be?

Do you have any explanation why GET-LT1 is your top performing model?

Does your model generalize to unseen molecules or reactions better than previous methods?

In Table 5, why does the % invalid SMILES go up with more beams? I would figure that larger beam sizes would result in more valid generated molecules?

In Table 4, please report the number of parameters for each model.

In Table 2, what are the astrices (*) in the GET-LT1 row?

In Table 3, you should bold the Similarity model for the top-5 and top-10 accuracy.

In equation 9, do you mean “sigmoid”?

In equation 8, should the k superscript be a subscript?

How would this model perform with SELFIES representation of small molecules, which are more robust representation [Krenn et al., 2019]?


**Experience Assessment:**

I have published one or two papers in this area.

**Review Assessment: Checking Correctness Of Derivations And Theory:**

I assessed the sensibility of the derivations and theory.

**Review Assessment: Checking Correctness Of Experiments:**

I carefully checked the experiments.

**Review Assessment: Thoroughness In Paper Reading:**

I read the paper thoroughly.

---

> ### Author Response · Authors · 2019-11-12
> **Author Response #3**
>
> Thanks for your constructive suggestions and questions. We appreciate your approval of the meaning of our work. Our replies are as follow:
> 1. Sorry for not providing the error bar in the original paper.  Here we provide the error bar of GET-LT1 (We experiment 3 times with different random seeds (41, 42, 43) ).
>
> Model  | top-1 | top-3 | top-5 | top-10
> ---------------------------------------------------------------------------------------------
> GET-LT1 (our) |  59.1±0.062 | 73.4±0.276 | 76.4±0.187 |  78.7±0.179
>
> The above is our current  result. We ensemble 10 GET-LT1 models with different parameters by averaging the models’ probability vectors, i.e., the output vector of the decoder, to decide the generated SMILES character at each time step. By this way, we further improve the accuracy.
>
> 2. Yes, bootstrapping Transformer model with multiple non-canonical SMILES for the same input molecules can improve performance. Someone has done this experiment. (It can be find in  https://github.com/kheyer/Retrosynthesis-Prediction) . With multiple valid SMILES, the model can learn the grammar of SMILES more easily and achieve better results.
>
> 4. Compared with GET-LT2, GET-LT1 utilizes the atom information;  Compared with GET-LG, GET-LT1's learned representations are more determined by the Transformer and we think this is why it is better.  Since the decoder is a "sequence decoder", it may be better to give the "sequential encoder"(i.e., Transformer encoder) more weights while using the graphical encoder (i.e., Graph Encoder) as an enhancement. Thus, letting the representations first pass through the graph encoder and  then pass through the Transformer encoder may be better; Compared with GET-CT,  fusing two-level's embeddings in a serial way while not in a parallel way (i.e., concatenation) can get better representations.
>
> 5. Yes, for the USPTO-50K dataset, our model's generative ability is the best.
>
> 6. We can get the top-N results with N beam size. As N become larger, the model's generative confidence for the top-N's result decrease, i.e., top-1 > top-2 > top-3 > top-4 > top-5. Larger N will result in more molecules which are generated with less confidence, so the invalid rate will go up.
>
> 7. We provide the concrete parameters of each model in our code. https://github.com/papercodekl/ICLR2020_MGET.  The training script is
> python -u train.py -data data2/seqdata -save_model experiments/checkpoints2/model -seed 42  -gpu_ranks 0 -save_checkpoint_steps 10000 -keep_checkpoint 20 -train_steps 500000 -param_init 0  -param_init_glorot -max_generator_batches 32 -batch_size 4096 -batch_type tokens -normalization tokens -max_grad_norm 0  -accum_count 4 -optim adam -adam_beta1 0.9 -adam_beta2 0.998 -decay_method noam -warmup_steps 8000  -learning_rate 2 -label_smoothing 0.0 -report_every 1000 -layers 4 -rnn_size 256 -word_vec_size 256 -encoder_type transformer -decoder_type transformer -dropout 0.1 -position_encoding -share_embeddings -global_attention general -global_attention_function softmax -self_attn_type scaled-dot -heads 8 -transformer_ff 2048.
>
> And we will supplement the parameter settings of each model in the paper later.
>
> 8.  The  astrices (*) means the best result among template-free methods.
> 9. Sorry, its our negligence.
> 10. Yes, I mean sigmoid, sorry for my spelling mistake.
> 11. k means the k-th head. We refer to GAT (Velickovic et al. 2018´) to write this symbol. Maybe it's better to write as a subscript.
>
> 12. SELFIES is a new proposed representation, thank you very much for telling us this informative paper and we will test our model with SELFIES later.

---

### Official Review · AnonReviewer1 · 2019-10-27
**Official Blind Review #1**

**Rating:** 1

**Review:**

This paper proposed Graph Enhanced Transformer(GET) to combine the graphical and sequential representations of the molecule to improve the retrosynthesis prediction performance. Experiments indicated that the proposed model outperforms state-of-the-art Seq2Seq-based methods on USPTO-50K dataset, and showed ability in reducing invalid SMILES rate.

Two main comments:

1. This paper provide no novelty with respect to deep learning method. It is just a combination of sequence transformer and graph neural network (using RDKit(Landrum, 2016) to transform a SMILES into the molecular graph). The decoder is the same as vanilla Transformer to generate SMILE string output.

2. The writing can be improved. For instance, in the caption of Figure 1 - "somehow to be transformed" ...  plus a few other places have wording issues like this.






**Experience Assessment:**

I have published in this field for several years.

**Review Assessment: Checking Correctness Of Derivations And Theory:**

I assessed the sensibility of the derivations and theory.

**Review Assessment: Checking Correctness Of Experiments:**

I assessed the sensibility of the experiments.

**Review Assessment: Thoroughness In Paper Reading:**

I read the paper at least twice and used my best judgement in assessing the paper.

---

> ### Author Response · Authors · 2019-11-12
> **Author Response #1**
>
> We thank the reviewer for the feedback and comments. Our replies are as follow.
>
> -"This paper provide no novelty with respect to deep learning method. It is just a combination of sequence transformer and graph neural network "
>
> Our work is the first attempt to fuse the graphical and sequential information of molecules to solve the retrosynthesis prediction task. We take the application of deep learning to retrosynthesis in novel directions.  We fully investigate four organization forms of graph neural network (GNN) with Transformer and significantly improve the prediction performance in many metrics, i.e.,  accuracy and chemical invalid rate.  Besides, the GNN we used is also well-designed to further improve the performance, which is more powerful to learn the representation of atoms.
> We note that the successful combination of GNN and Transformer model is meaningful since the molecule has graph structure naturally and sequential structure artificially. We think that how to jointly utilize these two forms' information effectively is important in this area, while it has not been explored in literature.  Our work just fills this gap and achieves good results.

---

### Decision · Program_Chairs · 2019-12-19

**Decision:**

Reject

**Comment:**

Several approaches can be used to feed structured data to a neural network, such as convolutions or recurrent network. This paper proposes to combine both roads, by presenting molecular structures to the network using both their graph structured and a serialized representation (SMILES), that are processed by a framework combining the strenth of Graph Neural Network and the sequential transformer architecture.

The technical quality of the paper seems good, with R1 commenting on the performance relative to SOTA seq2seq based methods and R3 commenting on the benefits of using more plausible constraints. The problem of using data with complex structure is highly relevant for ICLR.

However, the novelty was deemed on the low side. As a very competitive conference, this is one of the key aspects necessary for successful ICLR papers. All reviewers agree that the novelty is too low for the current (high) bar of ICLR.